# Natural Compounds, Optimal Combination of Brusatol and Polydatin Promote Anti-Tumor Effect in Breast Cancer by Targeting Nrf2 Signaling Pathway

**DOI:** 10.3390/ijms24098265

**Published:** 2023-05-05

**Authors:** Jing Li, Jianchao Zhang, Yan Zhu, Lukman O. Afolabi, Liang Chen, Xuesong Feng

**Affiliations:** 1Shenzhen Laboratory of Tumor Cell Biology, Center for Protein and Cell-Based Drugs, Institute of Biomedicine and Biotechnology, Shenzhen Institute of Advanced Technology, Chinese Academy of Sciences, Shenzhen 518055, China; ljysa1997@163.com (J.L.); yan.zhu@siat.ac.cn (Y.Z.);; 2School of Pharmacy, China Medical University, Shenyang 110122, China; 3Department of Biochemistry, School of Medicine, Southern University of Science and Technology, Shenzhen 518055, China; zhangjc3@sustech.edu.cn

**Keywords:** breast cancer, brusatol, polydatin, Nrf2, HO-1, NQO1, ROS

## Abstract

Triple-negative breast cancer (TNBC) has been clearly recognized as a heterogeneous tumor with the worst prognosis among the subtypes of breast cancer (BC). The advent and application of current small-molecule drugs for treating TNBC, as well as other novel inhibitors, among others, have made treatment options for TNBC more selective. However, there are still problems, such as poor patient tolerance, large administration doses, high dosing frequency, and toxic side effects, necessitating the development of more efficient and less toxic treatment strategies. High expression of Nrf2, a vital antioxidant transcription factor, often promotes tumor progression, and it is also one of the most effective targets in BC therapy. We found that in MDA-MB-231 cells and SUM159 cells, brusatol (BRU) combined with polydatin (PD) could significantly inhibit cell proliferation in vitro, significantly downregulate the expression of Nrf2 protein as well as the expression of downstream related target genes *Heme Oxygenase-1 (HO-1)* and *NAD(P)H dehydrogenase, quinone 1 (NQO1)*, and promote reactive oxygen species (ROS) levels to further strengthen the anti-tumor effect. Furthermore, we discovered in our in vivo experiments that by reducing the drug dosage three times, we could significantly reduce tumor cell growth while avoiding toxic side effects, providing a treatment method with greater clinical application value for TNBC treatment.

## 1. Introduction

Breast cancer (BC), as a relatively common cancer, has gradually become one of the important causes of cancer incidence and mortality in women worldwide [1,2]. Triple-negative breast cancer (TNBC) is a highly aggressive and heterogeneous subtype of BC, and has higher mortality, recurrence and metastasis rates with shorter overall survival (OS) [3]. Currently, surgery, radiotherapy, and chemotherapy are still the main options for TNBC treatment [4]. The emergence and application of some novel cancer treatment methods, such as immunotherapy, targeted therapy, especially small molecule targeted agents, have been emerging in recent years and have greatly improved the survival rate of advanced patients who do not respond well to chemoradiotherapy [5,6]. However, they do have some drawbacks, such as systemic toxicity, drug resistance, and low patient response rates [7,8,9]. Therefore, more efficient and less toxic methods of treating TNBC must be developed.

Nuclear factor erythroid 2-related factor 2 (Nrf2) is a nuclear transcription factor that can regulate a variety of antioxidant gene expressions in cells [7]. Under normal conditions, Nrf2 is inhibited by Keap1 and stably exists in the cytoplasm [10]. However, when subjected to oxidative stress, Nrf2 escapes repression by Keap1, translocates to the nucleus, and activates the expression of detoxification enzyme genes mediated by antioxidant response element (ARE), such as *Heme Oxygenase-1 (HO-1)* and *NAD(P)H dehydrogenase*, *quinone 1 (NQO1)* [11,12]. Currently, it has been confirmed that aberrant expression of the Nrf2 signaling pathway plays a key role in the development and progression of BC, invasion and metastasis, and therapeutic prognosis [13]. For example, it was recently shown that silencing Nrf2 significantly inhibited the proliferation and migration of MDA-MB-231 or MCF7 cells and that Nrf2 re-expression promoted cell growth and invasiveness [11,14,15,16]. Meanwhile, the clinical research data from BC patients revealed yet again that patients with a low survival rate and poor prognosis usually have increased *Nrf2* gene activity which promotes tumor cell proliferation, invasion and migration [17]. Consequently, targeting the Nrf2 signaling pathway may improve BC therapeutic efficacy.

TNBC treatment remains a significant clinical challenge to this day. Due to advantages such as target specificity and high selectivity, some candidate small molecule drugs, including single targeting drugs and repurposing drugs, have emerged as a promising therapy for TNBC [18]. In recent years, the FDA has approved small-molecule representative drugs for treating BC, including everolimus, abemaciclib, lapatinib, olaparib, talazoparib, fluoxysterone, etc. [19]. However, after a long period of treatment, there is usually a reduction of therapeutic efficacy and the development of drug resistance. For many years, Chinese herbal medicines have been used to treat cancer. Natural small molecule compounds extracted from them have gradually attracted much attention in BC treatment because of the advantages of high safety, rapid drug efficacy, non-toxic side effects, and reduced chemoresistance [20]. According to the current studies, a variety of natural small molecule compounds, such as bioflavonoids [21,22], resveratrol [23,24], brusatol [5,6], 10-gingerol [25,26], and others, have shown good anticancer activity against BC. Brusatol (BRU), a quassinoid, isolated from *Brucea javanica*, has been identified as a potent inhibitor of the Nrf2 signaling pathway [27,28]. In recent years, much evidence has demonstrated that BRU has good anti-tumor activity in different cancers, such as colorectal [29], liver [30,31], lung [32], and pancreatic cancer [33], as well as BC [34]. For example, Lu et al. reported that BRU could effectively inhibit the cell viability of colorectal cancer cells HCT116 [35]. Ren et al. found that BRU can inhibit the expression of c-myc in leukemia cells, which can increase the cytotoxicity, and inhibit the expression of Nrf2 from improving the therapeutic effect of chemoradiotherapy [36]. Polydatin (PD), extracted from the roots of *Polygonum cuspidatum*, is a resveratrol glycoside with anti-tumor, antioxidant and other biological activities [37,38]. Current studies have confirmed the therapeutic efficacy of PD in multiple cancers. For example, Zhang et al. found that PD could inhibit lung cancer cells A549 and NCI-H1975 proliferation and induce apoptosis [39]. Chen et al. found that PD could down-regulate the phosphorylation level of Creb and induce cell apoptosis in MDA-MB-231 and MCF-7 BC cells [40]. However, there are numerous disadvantages to using a single drug to treat cancer, including multiple administration times, large dosage, high cost, and a lengthy treatment cycle [10,20,29,37,41,42,43]. Therefore, combination strategies are being developed gradually in order to further optimize monotherapy and improve the survival rate of BC patients [44]. Recently, Chen et al. demonstrated that the synergistic effect of BRU with cisplatin could enhance the antitumor effect on colorectal cancer cells [45]. Xiang et al. found that BRU enhanced the anti-tumor effect of gemcitabine by inhibiting the Nrf2 signaling pathway in pancreatic cancer cells [33]. In addition to combination therapy strategies for PD, Zhang et al. first showed that combining PD with an inhibitory glycolytic compound, 2-Deoxy-d-glucose (2-DG), inhibited proliferation, migration and invasion and induced apoptosis in MCF7 and 4T1 cells [38].

In this study, we demonstrated that combining PD with BRU could improve TNBC anti-tumor efficacy by using a therapeutic regimen consisting of half the drug concentration and fewer administrations. The approach mechanistically inhibited the proliferation of BC cells MDA-MB-231 in vitro, downregulated the expression of Nrf2 and its downstream genes *HO-1* and *NQO1*, and promoted ROS production and accumulation in these tumor cells. Meanwhile, animal models also verified that low-dose BRU combined with low-dose PD could effectively slow down tumor growth under fewer administration times, which provided a new direction for BC treatment.

## 2. Results

### 2.1. BRU Combination with PD Inhibits TNBC Cells Proliferation and Significantly Increases ROS Levels In Vitro

To explore whether combining BRU with PD could improve anti-tumor efficacy, we first measured the cell viability in human TNBC cells MDA-MB-231 and SUM159 cells by CCK-8 analysis to determine the effect of drug synergy on cell proliferation in vitro. In addition, we also observed the effect of BRU combined with PD treatment on the proliferation of MDA-MB-231 cells through Flow cytometry. The CCK-8 results showed that treatment with the combination of BRU and PD at different treatment times was able to inhibit cell proliferation compared with the control group significantly, but the inhibition of cells proliferation by the treatment alone was lower than BRU+PD group (Figure 1A,B). The Flow cytometry results showed that the number of cell granules in the BRU+PD group was significantly less than the control group or the alone treatment group in the same volume. The cell proliferation rate was slow as the treatment time extended (Figure 1C).

It is well established that ROS produced in cancer is related to antioxidation, and has toxic effects on tumor cells at high levels [46,47]. We speculate that the inhibition of BRU and PD treatment on cell proliferation may be caused by the change in ROS levels. Flow cytometry was thus used to determine whether the co-treatment of BRU and PD would contribute to an increase in ROS accumulation. The results showed that, when compared to the control group both BRU and PD increased ROS levels after drug treatment alone, but the combined group of BRU+PD significantly promoted ROS levels (Figure 1D). The ROS levels of cells treated with BRU and PD alone increased by about 2-fold compared with the control group, while the ROS levels of cells treated with BRU+PD increased by about 2.5-fold (Figure 1E).

### 2.2. Combined Treatment of BRU and PD Downregulates Nrf2 Protein Expressions in TNBC Cells

Next, we explored the anti-tumor mechanism of BRU combined with PD in TNBC cells. Since the antioxidant system regulates the levels of intracellular ROS, and Nrf2 is one of the classical pathways in the antioxidant pathway, excessive activation usually promotes tumor cell survival. Hence, we further evaluated the expression of Nrf2 protein after PD combined with BRU treatment. We observed by immunofluorescence (IF) that after H_2_O_2_ stimulation of the cells, Nrf2 was inhibited from transferring to the nucleus after treatment with BRU and PD drugs. Nrf2 protein expression was significantly decreased in the BRU+PD group compared with the vehicle or other groups (Figure 2A). In addition, we also detected the total Nrf2 protein levels after the combined treatment of BRU with PD in MDA-MB-231 cells and SUM159 cells by Western blot method, as well as the change of Nrf2 protein levels after separating the nucleus from cytoplasm, the results of which were consistent with the IF results. It was reconfirmed that treatment with BRU combined with PD could more significantly inhibit Nrf2 protein expression than the control group (Figure 2B–G). Taken together, the anti-tumor effect of BRU+PD was through downregulating Nrf2 protein expression, which threatened the survival of TNBC cells.

### 2.3. The Combination of BRU and PD Downregulates the Expression of Nrf2 Downstream Target Genes HO-1 and NQO1

Studies have shown that Nrf2 can regulate the expression of target genes by binding to the antioxidant response element (ARE) of a series of antioxidant genes, such as *NAD(P)H: quinine oxidoreductase-1 (NQO1)* and *hemeoxygenase-1 (HO-1)* [6,48,49]. We hypothesized that the synergistic treatment of MDA-MB-231 cells with BRU+PD would also inhibit target genes downstream of *Nrf2*. To verify this speculation, we measured the changes in mRNA levels of *HO-1*, and *NQO1*, two target genes of the Nrf2 downstream pathway. The results indicated that compared to the control group, the expression of *HO-1* and *NQO1* could be downregulated by using BRU or PD alone in the presence of oxidative damage caused by H_2_O_2_ treatment. However, after combined therapy with BRU+PD, the down-regulation of *HO-1* and *NQO1* was significantly lower than that of drug treatment alone (Figure 3). In conclusion, these findings suggest that the anti-tumor mechanism of BRU and PD combined therapy inhibits Nrf2/HO-1 and NQO1 signaling pathways while increasing ROS levels, which has a toxic effect on BC cells and inhibits cell proliferation.

### 2.4. PD Combined with BRU Treatment Significantly Inhibited Tumor Growth In Vivo

Since combination therapy of PD with BRU had a better effect on inhibiting cell proliferation in tumor cells, we further evaluated the anti-tumor effect of combination therapy in vivo. Here, we used a therapeutic strategy at a concentration of 25 mg/kg of PD synergistically with 1 mg/kg of BRU and administered it only three times; MDA-MB-231 cells were subcutaneously inoculated into nude mice to construct a mouse model (Figure 4A). Nude mice with established xenograft tumors were randomized, and i.p. administration was started when tumors grew to 100–200 mm^3^. According to the results, the tumors in the BRU+PD group were significantly smaller than those in control versus PD groups (Figure 4B,C), and the combination of PD and BRU resulted in about 50% reduction in the weight of the tumors as well (Figure 4D). In addition, we also evaluated the nonspecific toxicity of PD synergistic BRU combinational treatment in transplanted tumors. The results showed that PD combined with BRU could effectively inhibit tumor growth but did not lead to significant weight loss in mice (Figure 4E). Taken together, these findings suggest that BC patients may benefit from the combination treatment strategy of BRU+PD and significantly reduce the dosage and the number of administrations, which is expected to improve patient clinical outcomes.

## 3. Discussion

As a common subtype of BC, TNBC remains one of the most prevalent malignant tumors threatening women’s health worldwide. Although chemotherapy is the only treatment method currently available, the treatment effect is still poor due to tumors aggressive invasion, poor prognosis, and drug resistance [1,2]. For many years, Chinese herbal medicine has been used to treat cancer. Due to its benefits, including low toxicity and side effects, reversing drug resistance, it is gaining increasing attention. As herbal extracts, BRU and PD, have shown many biological activities, especially good anti-tumor activity. Previously, many studies have shown that BRU combined with other drugs showed good anti-breast cancer effects. For instance, in the in vitro study of Chandrasekaran et al. found that the combination of BRU with paclitaxel showed synergistic anti-tumor effects and effectively inhibited the migration and invasion of TNBC cells [5]. Tian and Yang et al., respectively, found that the Nrf2 inhibitor BRU exerted synergistic effects and targeted Nrf2 related pathways for treating HER2 positive tumors in combination with lapatinib and trastuzumab, respectively [6,50]. Therefore, BRU has been proven to be a natural and effective anti-tumor or adjuvant anti-cancer drug with significant advantages. At present, some studies also have shown the therapeutic effect of PD on TNBC, such as Liu et al. found that PD can induce pyroptosis in TNBC to exert anti-cancer effects [51]. The utilization of natural products for substitution shows much greater potential due to the limited efficacy of traditional chemotherapeutic drugs as well as the expensive cost, and lack of effective targeted drugs. Therefore, in this research, we have proved that the combination of BRU and PD can inhibit Nrf2/HO-1 and NQO1 pathway, produce ROS, increase its level, and inhibit the proliferation of TNBC cells. In animal models, based on previous studies, such as BRU at a concentration of 2 mg/kg and PD at a concentration of 50 or 100 mg/kg administered daily or every two days [31,37,42,43,50]. We optimized the treatment method, and inhibit tumor growth by reducing the dosage and times of administration, which shows the therapeutic advantage of BRU+PD in TNBC treatment.

Under normal physiological conditions, the redox system in the body maintains a coordinated and balanced state. ROS can complete specific physiological functions such as signal transduction and protein regulation. However, due to ROS’s dual nature, high levels of ROS also significantly affect the tumor phenotype and growth during cancer development [52]. In this study, we found that the combination of BRU+PD significantly reduced the proliferative capacity of MDA-MB-231 cells and SUM159 cells relative to either treatment alone, and ROS levels were also significantly higher in the other groups. These results demonstrated that the potent antiproliferative effect of BRU+PD on TNBC cells in vitro may be due to the elevation of ROS. However, there is also debate regarding the role of ROS in tumor progression. Since either extremely high or extremely low levels of ROS can affect the changes in ROS-mediated signaling pathways, therapies that eliminate ROS or enhance ROS production are all potential cancer treatments [53]. Therefore, we hypothesized that the combination of PD and BRU may also affect ROS levels and thus tumor cell growth through other signaling pathways. In addition, it is also worth exploring whether other natural small-molecule drugs with anti-tumor activity can achieve more significant anti-tumor effects when they are used in combination.

It has been shown that the Nrf2 transcription factor can protect normal cells from oxidative damage by directing the host to respond to intracellular oxidative stress by regulating a series of effectors. But the high Nrf2 expression can also protect tumor cells from chemotherapeutic damage [54]. Therefore, Nrf2 has been a key target in cancer therapy. In our study, it was found that the expression of Nrf2 in MDA-MB-231 cells could be downregulated after the synergistic treatment of PD with BRU, and the antioxidant response was inhibited, while downstream target genes such as *HO-1* and *NQO1* were downregulated. These findings indicate that the anti-tumor efficacy of BRU+PD is achieved by inhibiting the expression of Nrf2 as well as its downstream target genes, thereby increasing ROS levels leading to inhibiting tumor cell proliferation. However, it remains unclear whether the regulatory mechanism of PD on Nrf2 is a direct inhibition of Nrf2 activity, which still needs to be further verified. In addition, we now know that Nrf2 is widely recognized as an oxidative stress regulator, but numerous studies have shown that Nrf2 also has an essential role in cancer cell metabolism [55]. Activation of Nrf2 can participate in cancer cell metabolic processes and regulate multiple metabolism related key target genes, such as the production of NADPH, including the main enzymes catalyzing NADPH synthesis: G6PD, IDH1/IDH2, ME1, and PGD, as well as the regulation of the pentose phosphate pathway (PPP), and involved in the regulation of genes involved in glutathione metabolism and glutamine metabolism, among others [56,57]. According to Chio et al., combined targeting inhibition of Akt signaling and synthesis of glutathione could mimic Nrf2 ablation and effectively inhibit the survival of pancreatic ductal adenocarcinoma cells (PDACs) in vitro and in vivo [58]. In a study by Mukhopadhyay et al., increased Nrf2 levels in PDACs were found to be mainly due to altered pathways regulated by glutathione metabolism and demonstrated an important role for the treatment of PDACs by targeting glutathione metabolism [59]. While in the study on breast cancer, Zhang et al. showed that inhibition of Nrf2 can downregulate the expression of G6PD and pentose phosphate transports pathway and further affect the signaling of other pathways, thereby inhibiting the proliferation and migration of BC cells [60]. Based on the above findings all illustrate that in addition to the important role of targeting the Nrf2 antioxidant related pathway for the treatment of breast cancer, Nrf2 is a nonnegligible key player in breast cancer cell metabolism. And previous studies have confirmed that PD was identified as a potent inhibitor of G6PD [61], so we speculated that PD might also inversely regulate Nrf2 expression by regulating G6PD. In our subsequent research work, the effect of combination therapy of BRU with PD on the signaling pathways of downstream effectors of the Nrf2 signaling pathway in TNBC cells should also be further demonstrated.

## 4. Materials and Methods

### 4.1. Reagents and Cell Culture

Brusatol (HY-19543, purity 99.89) and polydatin (HY-N0120A, purity 98.55) were obtained from MedChemExpress (Monmouth Junction, NJ, USA). DMSO was obtained from Beyotime Biotechnology (Shanghai, China). CH_3_COOK, MgCl_2_, DTT, NaCl were fetched from Sangon Biotech (Shanghai, China). Triton X-100 was purchased from Aladdin (Shanghai, China). DMEM and RPMI-1640 medium were obtained from Biological Industries (Kibbutz Beit Haemek, IL, USA). Fetal bovine serum (FBS) was obtained from MIKX (Shenzhen, China). CCK-8 kit was obtained from Yeasen (Shanghai, China). The commercial antibodies used in the study: Nrf2 and LaminA/C were obtained from Cell Signaling Technology (Danvers, MA, USA). GAPDH is obtained from Proteintech (Wuhan, China). HRP-conjugated affinipure goat anti-rabbit/mouse IgG (H+L) secondary antibodies were obtained from Proteintech.

Human BC MDA-MB-231 cells and SUM159 cells were purchased from Procell Life Science & Technology Co., Ltd. (CL-0150B and CL-0622, Wuhan, China), MDA-MB-231 cells were cultured in DMEM with high glucose (BI) supplemented with 10% FBS and 1% of penicillin/streptomycin (Beyotime) at 37 °C and 5% CO_2_. SUM159 cells were cultured in RPMI-1640 medium (BI) supplemented with 10% FBS and 1% of penicillin/streptomycin (Beyotime) at 37 °C and 5% CO_2_.

### 4.2. Detection of Intracellular ROS

Intracellular ROS levels were measured using H2DCFH-DA (Molecular Probes, Invitrogen, Carlsbad, CA, USA). The specific procedure was performed as follows: MDA-MB-231 cells were incubated with or without BRU (20 nM) and PD (100 nM) for 24 h in the dark in the cell culture incubator. After being washed three times with PBS, 100 µM H_2_O_2_ was co-incubated with cells for 4–6 h; then washed with PBS, and treated with 10 µM H2DCFH-DA working solution for 20–30 min. Finally, the cells were collected with trypsin, washed twice, and then suspended in 500 µL PBS. DCF intensity was measured by flow cytometer at excitation/emission wavelengths of 488/525 nm.

### 4.3. CCK-8 Assay

In this experiment, Cell Counting Kit-8 (Yeasen, Shanghai, China) was used to estimate cell viability. MDA-MB-231 Cells and SUM159 cells were cultured in 96-well cell culture plates (5 × 10^3^ cells/well), then, 100 μL DMEM was added to each well. After the cellular adherence, cells were treated with 20 nM BRU and 100 nM PD alone or in combination for 24, 48 and 72 h. Before determination, we added 10 μL CCK-8 reagent to each well and incubated with cells at 37 °C; for 30 min–1 h. Then measure the absorbance of each hole at 450 nm with a microplate reader (Bio-Rad, Hercules, CA, USA). Finally, we calculated and analyzed the data using GraphPad Prism 9 (GraphPad Prism Software Inc., San Diego, CA, USA). The experiment was repeated three times.

### 4.4. Western Blot Assay

MDA-MB-231 cells and SUM159 cells were seeded in 6-well tissue culture plates (Corning Incorporated, Corning, NY, USA) and allowed to attach for 24 h. The cells were treated with drugs after the cell density was about 80–90%: 20 nM BRU and 100 nM PD alone or jointly treated the cells for 24 h, and then the cells were collected. After washing with PBS twice, SDS lysis buffer (Beyotime. Shanghai, China) was added to the well plates and waited for 1–3 s to completely lyse the cells. Then added 5 × sample loading buffer into the protein samples and boiled them for 10 min. The protein sample was electrophoretic on 10% SDS-PAGE prefabricated gel (Yeasen, Shanghai, China) and transferred onto a polyvinylidene fluoride (PVDF, Millipore, Bedford, MA, USA) membrane. The membranes were blocked with 5% non-fat milk for 1 h at room temperature (RT), followed by primary antibody incubation overnight at 4 °C. The membrane was washed using PBS (PBST) containing 0.1% Tween-20 (Sangon Biotech, Shanghai, China) for a total of 3 times, 5 min/time, and then the secondary antibody was incubated for 1 h at room temperature. Finally, the detection of bands was performed using hypersensitivity enhanced chemiluminescence (ECL, Yeasen, Shanghai, China) in an ultrasensitive multifunctional gel imager (Cytiva, Amersham Imager AI600, Washington, DC, USA). Analysis of band gray scale values was done using Image J software (NIH, Bethesda, MD, USA). The statistical analysis of data mainly used GraphPad Prism 9 (GraphPad Prism Software Inc., San Diego, CA, USA).

### 4.5. Nuclear and Cytoplasmic Extraction

MDA-MB-231 cells were incubated in a 6 cm petri dish (sterile) overnight and harvested after drug treatment, then cells were resuspended in buffer A (20 mM HEPES, 5 mM CH_3_COOK, 1 mM MgCl_2_, 0.5 mM DTT, pH 7.9) on ice for 30 min, and cells were repeatedly aspirated approximately 30 times by using the 30 G insulin syringe (Becton, Dickinson and Company, Franklin Lakes, NJ, USA). The cell lysates were first centrifuged at 4000 rpm for 5 min, and the supernatant was further centrifuged at 14,000 rpm for 40 min to gather the cytoplasmic fraction. Nuclear fraction proteins were next collected, and the pellet obtained by the first centrifugation was incubated with buffer B (20 mM HEPES, 5 mM CH_3_COOK, 1 mM MgCl_2_, 0.5 mM DTT, 0.4 M NaCl, pH 7.9) at 4 °C for 20 min and then centrifuged at 14,000 rpm for 40 min.

### 4.6. Immunofluorescence

Cells were adherent overnight in 24 well plates (Corning Incorporated, Corning, NY, USA) containing cell climbing pieces. When the cell density in the well plate reached about 60–70%, the GFP-Nrf2 plasmid (plasmids were obtained from the laboratory) was transfected into the cells. After 12 h of plasmid expression, cells were transfected using 100 μM H_2_O_2_ for 4–6 h, and then cells were treated using 20 nM BRU alone or in combination with 100 nM PD for 24 h. Finally, washed 1–2 times with PBS, and cells were fixed with 4% PFA for 15 min, followed by permeabilization with 0.1% Triton X-100 for 15 min. Then used PBS to wash cells three times for 5 min at a time. DAPI alone stained the nucleus for 5–10 min, and washed it with PBS three times, 5 min at a time. At last, the anti-fluorescence quencher agent containing DAPI (Vector Labs, Burlingame, CA, USA) was used to seal the film and observed under the confocal fluorescence microscope (Olympus, Tokyo, Japan).

### 4.7. Quantitative Real-Time PCR

Cells were collected after treated with 20 nM BRU and 100 nM PD alone or in combination for 24 h. Total RNA was extracted with TRIzol reagent (Invitrogen, Carlsbad, CA, USA). RNAs were converted to cDNAs using a reverse transcription Kit (Takara, Tokyo, Japan); using SYBR Green PCR Master Mix (2X) (Vazyme, Nanjing, China), mRNA expression was measured, and the mRNAs were quantified by ΔΔ Ct method, and GraphPad Prism 9 (GraphPad Prism Software Inc., San Diego, CA, USA) was used for data analysis.

The Q-PCR primers for *HO-1* were 5′-ACATCGACAGCCCCACCAAGTTCAA-3′ and 5′-CTGACGAAGTGACGCCATCTGTGAG-3′. The Q-PCR primers for *NQO1* were 5′-GGATTGGACCGAGCTGGAA-3′ and 5′-AATTGCAGTGAAGATGAAGGCAAC-3′.

### 4.8. Xenograft Models in Nude Mice

BALB/C nude mice (female, 5 weeks old) were purchased from Zhuhai BesTest Bio-Tech Co., Ltd (Zhuhai, China). BC cells MDA-MB-231 (100 μL containing 5 × 10^6^ cells) were injected subcutaneously into nude mice. Until the tumor volume reached 100–200 mm^3^, mice in the individual administration groups were injected intraperitoneally with PD 50 mg/kg, the combined administration group was PD 25 mg/kg and BRU 1 mg/kg. Mice in the control group were intraperitoneally injected with PBS (containing 1% DMSO) once every seven days for a total of three times. The mice were monitored for tumor growth, and tumor volume was measured every seven days, and finally, the mice were sacrificed, and the tumor tissues were taken and weighed. The tumor volume was calculated according to the formula (length × width^2^/2). All animal experiments were undertaken in accordance with relevant guidelines and regulations and were approved by the Institutional Animal Care and Use Committee at SIAT.

## 5. Conclusions

In conclusion, this study aims to determine a novel low-cost, low-dose, less frequent combination therapy strategy to enhance the anti-tumor effect of TNBC (Figure 5). Treatment with the Nrf2 inhibitor BRU in combination with PD downregulated the expression of Nrf2 and its downstream genes *HO-1* and *NQO1*, and promoted ROS production and accumulation, and inhibit MDA-MB-231 cells and SUM159 cells proliferation. This treatment strategy can greatly alleviate the treatment cost of BC patients and improve the therapeutic effect, with great application potential.

## Figures and Tables

**Figure 1 ijms-24-08265-f001:**
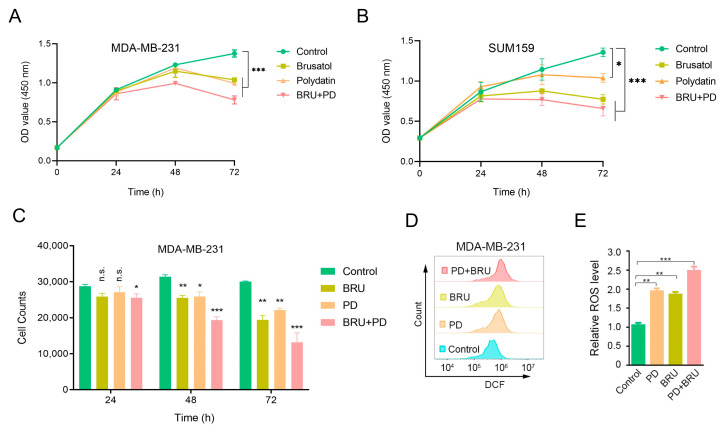
The synergistic effect of BRU and PD inhibited the proliferation of MDA-MB-231 cells and SUM159 cells in vitro and enhanced the reactive oxygen species (ROS) levels of MDA-MB-231 cells. (**A**,**B**) MDA-MB-231 cells and SUM159 cells were treated with BRU (20 nM) alone or in combination with PD (100 nM) for 24, 48 and 72 h, and then the cell viability was determined by the CCK-8 method. (**C**) After treating MDA-MB-231 cells with BRU (20 nM) and PD (100 nM) alone or in combination for 0, 24, 48, and 72 h, Flow cytometer was used to detect the changes in the number of cell proliferation granules of each group in the same volume (400 μL/tube) of cell solution. (**D**,**E**) Flow cytometer was used to detect the intracellular ROS levels change by fluorescent indicator H2DCFH-DA after treatment of cells with BRU (20 nM) and PD (100 nM) for 24 h. All results were expressed as the mean ± SD from three independent experiments and the statistical significance was evaluated by two-way ANOVA, n.s. not significant.* *p* < 0.05, ** *p* < 0.01, *** *p* < 0.001.

**Figure 2 ijms-24-08265-f002:**
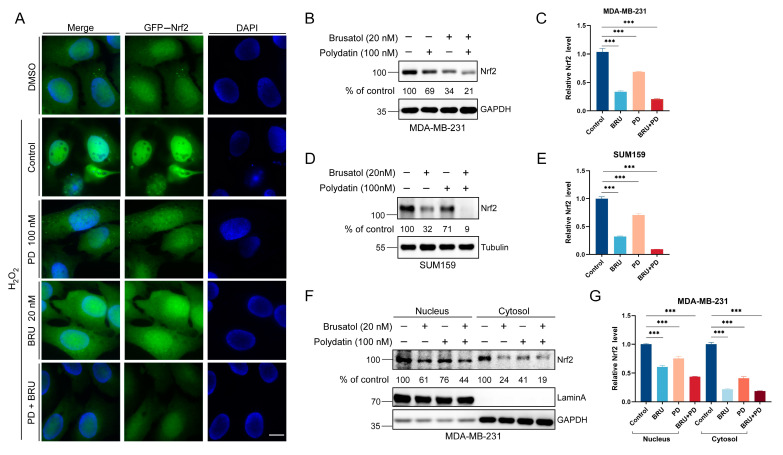
The effect of combining BRU with PD on Nrf2 protein expression in MDA-MB-231 cells and SUM159 cells. (**A**) The GFP-Nrf2 plasmid was transfected into MDA-MB-231 cells. After the plasmid was expressed for 12 h, it was treated with or without H_2_O_2_ 100 μM for 4–6 h. Then it was treated with 20 nM BRU alone or in combination with 100 nM PD for 24 h, immunofluorescence staining and observation was performed. Scale bar, 10 μm, (*n* = 3). (**B**,**C**) MDA-MB-231 cells were treated with indicated concentrations of BRU or PD for 24 h and total proteins were extracted for Western blot analysis. The bands were analyzed using the ImageJ software with a grayscale value analysis and statistical analysis by GraphPad Prism 9. (**D**,**E**) SUM159 cells were treated with indicated concentrations of BRU or PD alone or in combination for 24 h and total proteins were extracted for Western blot analysis. And bands were analyzed and quantified using ImageJ software and GraphPad Prism 9. (**F**,**G**) Nuclear/cytosolic fractionation was performed in MDA-MB-231 cells, and detected the changes of Nrf2 protein levels by Western blot assay, the bands were analyzed using the ImageJ software with a grayscale value analysis and statistical analysis by GraphPad Prism 9. Statistical significance was assessed using One–way ANOVA (*n* = 3). *** *p* < 0.001. “–” and “+” indicate whether or not corresponding drug treatments were performed.

**Figure 3 ijms-24-08265-f003:**
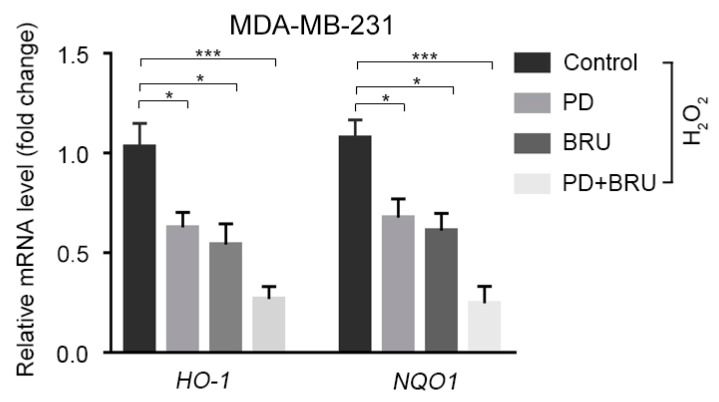
Relative mRNA level changes of Nrf2 downstream target genes *HO-1* and *NQO1* after BRU+PD treatment in MDA-MB-231 cells. Statistical significance was assessed using two-tailed Student’s *t*-tests (*n* = 3). * *p* < 0.05, *** *p* < 0.001.

**Figure 4 ijms-24-08265-f004:**
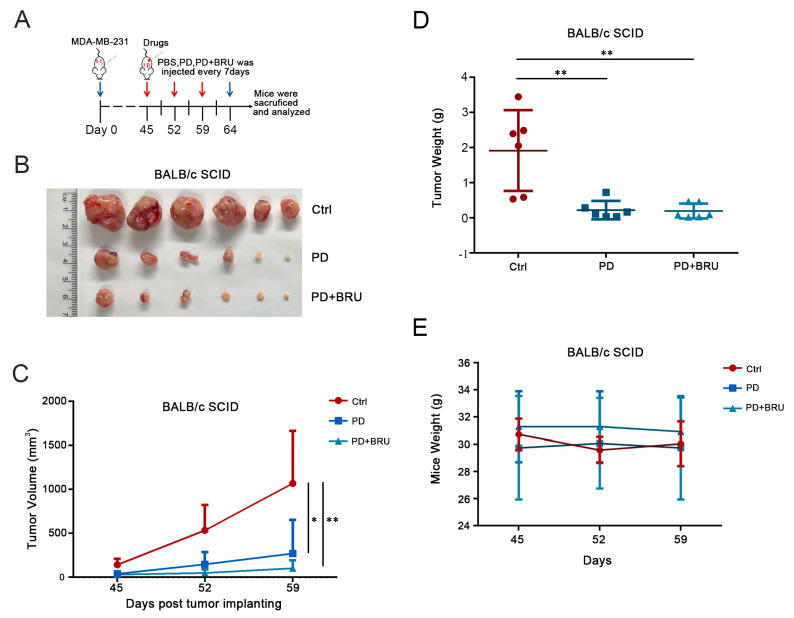
Inhibition of transplanted tumor growth in vivo by BRU and PD alone or in combination. BALB/c (female, 5 weeks old) were divided into three groups (*n* = 6) and subcutaneously injected with 5 × 10^6^ cells; tumor volume and mice weight were measured every 7 days and administered by i.p., while the control group received i.p. of PBS (containing 1% DMSO); Mice in the vehicle alone group were injected intraperitoneally with PD 50 mg/kg; The combination group received PD 25 mg/kg and BRU 1 mg/kg. (**A**) Schematic diagram of administration mode and treatment cycle of nude mice. The blue arrow indicates the starting date of subcutaneous tumor transplantation in mice and the dissection and analysis of the transplanted tumor on the 64th day. The red arrow indicates the position representing the time and frequency of each administration. (**B**,**D**) Tumors removed from mice after the last treatment was weighed. (**C**) Tumor volume was measured using vernier calipers and calculated using the following formula: V = length × width^2^/2 at the indicated time points. (**E**) Changes in body weight of mice during treatment. Data represent the mean ± SD, and the symbols * and ** denote significant differences of *p* < 0.05 and *p* < 0.01.

**Figure 5 ijms-24-08265-f005:**
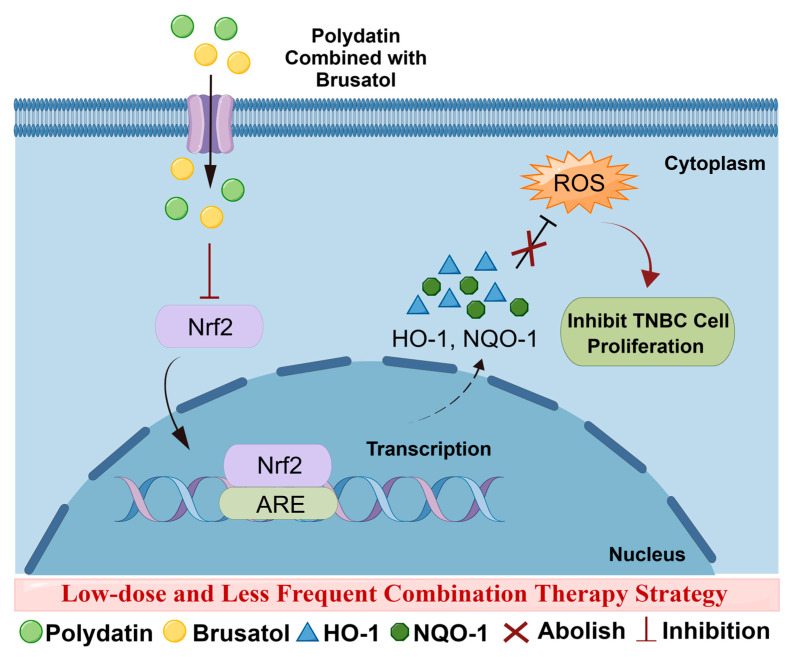
Scheme of combination therapy strategy for BRU and PD at low-dose and less frequent for TNBC treatment. This figure was drawn by Figdraw.

## Data Availability

Data are contained within the article.

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
