# Peer review of "Natural Compounds, Optimal Combination of Brusatol and Polydatin Promote Anti-Tumor Effect in Breast Cancer by Targeting Nrf2 Signaling Pathway"

_ijms, 2023, doi:10.3390/ijms24098265_

Round 1

Reviewer 1 Report

Generally, the paper is well-written and easy to follow. However, I suggest the results section should contain "results only" without other discussions and comparisons with the literature. These are dedicated to the "discussion" chapter.

Instead of the CCK-8 method used for quantification of the cells' viability, the authors could use flow cytometry, which is more precise and is available in their group.

ROS production is not compulsory a sign of induced cell apoptosis. Tumoral cells have mechanisms to resist these apparently deleterious effects. To sustain this assumption using flow cytometry is far more appropriate. 

Figure 2. I would expect to have an image of Nrf2 expression in "Control" without H2O2.  Please indicate what are the conditions for each column. Please send a supplementary figure with the whole WB blot.

The authors should argue why from the multitude of genes over or down-regulated in oxidative stress they choose only HO-1 and NQO1.

Brusatol was proven to inhibit nrf2 in mammalian cells (Olanyanju et al, 2015,) and also in Her2-positive breast cancer by part of the same group (Ziyin Tian, 2022). The same situation for polydatin, many previous studies show its efficacy in many cancers including breast cancer.  Please critically comment on the previous results and the novelty brought by this study. If the present study brings only the combination of these two drugs, more experiments are needed to prove their efficacy and convergent actions. Moreover, the title should stress that the combination is essential in this study and in fact the only novelty addressed.

Reviewer 2 Report

Article by Dr. Chen and the group elaborating on the role of Brusatol and Polydatin as the inhibitor of NRF2-regulated pathways with less toxic side effects, thus providing a treatment method with greater clinical application value for TNBC treatment. Though the novel combination therapy is promising with the translational aspect, few things must be addressed before it is ready for acceptance; they are as follows:

1. Authors must use another cell line to show the similar effect of the combination therapy. Utilizing one cell line is not sufficient for the publication with IJMS standards.

2. Authors must show the effect of the combination therapy with the signaling aspect of the downstream effectors of NRF2 signaling pathways with the metabolism aspect. At least discuss those as one of the future studies to be followed up further. Please refer to PMID: 31911550 and PMID: 27477511. 

3. Authors should add a model summarizing the findings of this manuscript with the known facts in the field. 

Round 2

Reviewer 1 Report

The authors accepted all my suggestions and performed the changes I considered to improve the MS. I recommend accepting this form.

Reviewer 2 Report

All concerns have been addressed, ready for acceptance.